

# Syntax and prejudice: ethically-charged biases of a syntax-based hate speech recognizer unveiled

Michele Mastromattei[1], Leonardo Ranaldi[2], Francesca Fallucchi[2] and Fabio Massimo Zanzotto[1]

[1] Department of Enterprise Engineering, University of Roma "Tor Vergata", Rome, Italy
[2] Department of Innovation and Information Engineering, Guglielmo Marconi University, Rome, Italy

## ABSTRACT

Hate speech recognizers (HSRs) can be the panacea for containing hate in social media or can result in the biggest form of prejudice-based censorship hindering people to express their true selves. In this paper, we hypothesized how massive use of syntax can reduce the prejudice effect in HSRs. To explore this hypothesis, we propose Unintended-bias Visualizer based on Kermit modeling (*KERM-HATE*): a syntax-based HSR, which is endowed with syntax heat parse trees used as a post-hoc explanation of classifications. KERM-HATE significantly outperforms BERT-based, RoBERTa-based and XLNet-based HSR on standard datasets. Surprisingly this result is not sufficient. In fact, the post-hoc analysis on novel datasets on recent divisive topics shows that even KERM-HATE carries the prejudice distilled from the initial corpus. Therefore, although tests on standard datasets may show higher performance, syntax alone cannot drive the "attention" of HSRs to ethically-unbiased features.

## INTRODUCTION

Hate speech has boomed with the use of social media and can turn out to be their ruin if not correctly moderated. Anonymity promotes hate to spread in online discussion (*Erjavec & Kovačič, 2012*). The term is so frequently used that seems to be crystal clear. Yet, the phenomenon of hate speech may result to be more difficult to capture than expected. Indeed, the major social networks and public entities give different definitions of it (*Fortuna & Nunes, 2018*) and the boundary among hate speech comments and similar concepts (such as *offensiveness, toxicity, aggressiveness, etc.*) is not obvious (*Poletto et al., 2021*). Building automatic hate speech recognizers is then a very hard challenge.

Hate speech recognizers (HSRs) (*Warner & Hirschberg, 2012*; *Djuric et al., 2015*; *Gambäck & Sikdar, 2017*) offer a tremendous opportunity to calm people down in "*social media arena*" (*Kirti & Karahan, 2011*). HSRs aim to recognize posts or comments which are recognized as full of hate. These posts or comments may be hidden and, possibly, related accounts may be blocked. This may contribute to control the behavior of individuals and can preserve online communities. Indeed, censoring hateful comments is not a limitation

Corresponding author
Michele Mastromattei,
michele.mastromattei@uniroma2.it

on free speech but, on the contrary, *"free speech"* is where communities (ethnic, gender, *etc.*) are preserved (*West, 2012*). Precisely on "free speech" and its meaning, in recent years a new form of censorship is being created—especially on social networks—based solely on the training data entered into HSRs. This new censorship can be guided by prejudice. In fact, prejudice can be injected into automatic recognizers by ethically-charged biases arising from learning data (*Carpenter, 2015*; *Crawford, 2016*; *Isaac, 2016*). Ethically-charged biases are extremely more dangerous than unintended/non-causal biases, which can lead to a conclusion utilizing wrong premises (*Yapo & Weiss, 2018*).

Word-based and transformer-based models are prone to include prejudice in hate speech recognizers. In these models, words tend to have a predominant role and, then, guide the final decision (*Burnap & Williams, 2015*; *Kwok & Wang, 2013*). However, words are often misinterpreted and their sole presence is used to determine whether comments or posts have abusive or offensive language. To mitigate this phenomenon, there are regularization solutions that eliminate any bias by introducing ad-hoc words in contexts (*Kennedy et al., 2020*) in order to better model the attention mechanism in transformers. As attention seems to capture syntactic information (*Eriguchi, Hashimoto & Tsuruoka, 2016*; *Chen et al., 2018*; *Strubell et al., 2018*; *Clark et al., 2019*), this is a way to start to include syntactic/structural information in the decision process. However, even if it is not clear how these regularizers reduce the use of trigger words, syntactic/structural information over sentences is used as a way to focus learning on features which are less biased.

In this paper, we push forward the research on how syntactic information can be used to de-bias hate speech recognizers and, thus, contribute to solve problems of prejudice. We then propose Unintended-bias Visualizer based on Kermit modeling (KERM-HATE): a Hate Speech Recognizer based on KERMIT (*Zanzotto et al., 2020*).

KERM-HATE offers two important features:

1. It includes syntactic trees as part of the architecture.
2. It offers a way of visualizing activation of syntactic trees as *post-hoc* (*Hase & Bansal, 2020*) explanation of decisions.

Syntactic trees should focus decisions on structural features, which are naturally ethically-unbiased (see Fig. 1). Experiments show that KERM-HATE significantly outperforms BERT-based hate speech recognizers on standard datasets (*Waseem & Hovy, 2016*; *Davidson et al., 2017*; *Founta et al., 2018*). However, on newly collected out-of-domain social media posts, KERM-HATE shows the limitation of the combined use of an ethically-biased dataset and a learning algorithm. Our experiments showed that even syntax-based models absorb prejudice from data. In fact, the manual analysis of post-hoc explanations of decisions shows that KERM-HATE, learned on an existing dataset (*Davidson et al., 2017*), is probabilistic and ethically biased.

## BACKGROUND AND RELATED WORK

Statistical bias is a fairly known problem in machine learning: *statistical bias* is the systematic, residual error that a learned model is expected to make when trained on a finite training set (*Dietterich & Kong, 1995*). This definition focuses only on performances

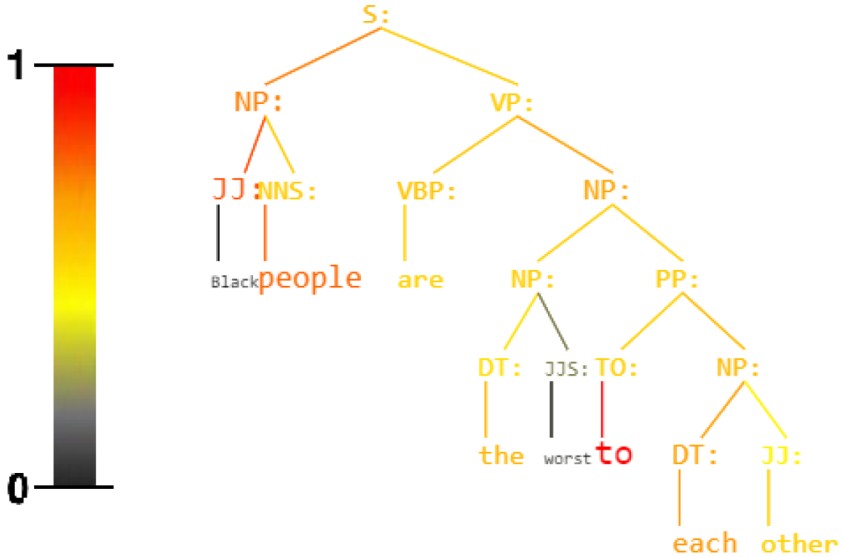

**Figure 1** **Sentence: Black people are the worst to each other.** Unbiased Syntax Heat Parse Tree derived by KERMIT (*Zanzotto et al., 2020*) within an Hate Speech Recognizer trained on the Davidson Corpus (*Davidson et al., 2017*). Active nodes are red.

of a model and, then, in principle is strictly correlated to its accuracy when deployed. Hence, it is not an ethical problem but something strictly correlated with the imprecise nature of learned models.

Statistical bias becomes an ethical issue if residual errors of a learned model occur more often for linguistic productions of specific social groups. In this case, statistical biases become *prejudice*, as if it is a "*mental state*" of the model with respect to social groups justified by stereotyped believes (*Quasthoff, 1989*). As in the general probabilistic bias, learning algorithms absorb prejudice from training corpora (*Caliskan, Bryson & Narayanan, 2017*), which generally contain stereotypes (*Stubbs, 1996*). This kind of bias is not only related with stereotypes, but also gender: there are numerous studies that have shown this in NLP applications (*Font & Costa-Jussa, 2019*; *Vanmassenhove, Hardmeier & Way, 2019*; *Lu et al., 2020*) and inside neural networks (*Zhao et al., 2019*).

In Hate Speech Recognition, *Sap et al. (2019)* conducted a case study regarding racial bias and specifically how the slang used predominantly by African-Americans—called AAE (African American English)—turns out to be more offensive than non-AAE equivalents—called SAE (Stanford American English)—relating the same phrase. They showed how same models propagate racial biases because they were trained on corpora that contained them.

Determining if and how much a learned model has prejudice is then a key important social issue. To solve this problem, operative definitions of prejudice in learned models have emerged. One of this is *unintended bias*: a model has *unintended bias "if it performs better for comments containing same particular identity terms than for comments containing others"* (*Dixon et al., 2018*). Since this definition is linked to particular identity terms, it

gives the basis to define a measure for determining prejudice of learned models. Therefore, *Dixon et al. (2018)* have introduced a measure for unintended bias called *Pinned AUC*: **P**inned **A**rea **U**nder the **C**urve to evaluate and compare unintended bias in trained models. However, this measure seems to be inefficient when datasets used as testing are unevenly distributed across different social groups (*Borkan et al., 2019*) .

Unfortunately, the above measures of prejudice for learned models fail to capture another important fact: learned models can be *"right from the wrong reasons"* (*McCoy, Pavlick & Linzen, 2019*; *Kamishima et al., 2012*). This is an important form of prejudice that should be unveiled too.

Explainable Artificial Intelligence (XAI) (*Adadi & Berrada, 2018*; *Hase & Bansal, 2020*), and explainable neural networks may help in determining whether learned models give the right or the wrong answers relying on prejudice. Indeed, explaining how machine learned models take their decisions is surging. This challenge is becoming a clear scientific endeavor (*Doshi-Velez & Kim, 2017*). Interesting XAI are mainly applied to the obscure neural networks (*Kahng et al., 2017*; *Vig, 2019*). In particular, KERMIT (*Zanzotto et al., 2020*) offers the possibility to explore how syntactic information is used in the decision process of a neural network. Hence, it is particularly useful in our study and may help in shedding light on how unintended bias or prejudice is absorbed from training data and persists in the final learned model.

## METHODS AND DATA

To explore our hunch that syntactic interpretation may help in de-biasing hate speech recognizers, we definitely need:

1. A Hate Speech Recognizer, which is based on syntactic interpretation and has the possibility to explain its decisions ('An Explainable Syntax-based Hate Speech Recognizer' section).
2. A definition of what prejudice in machine learning is ('Prejudice in Machine Learning Models' section).
3. A hate speech training corpus built to reduce the ethically-charged or, at least, the lexical probabilistic bias ('Possibly Unbiased Training and Validation Corpus' section)
4. Some fresh datasets on divisive topics, uncorrelated with the one selected for training ('Challenging Hate Speech Recognizers on Hot Topics' section).

First of all, we give a clear and effective definition of *"hate speech"* used as a basis throughout the paper. For this purpose, we use the definition employed by the *Committee of Ministers of the Council of Europe (1997)* that we quote below:

**Hate speech** shall be understood as covering all forms of expression which spread, incite, promote or justify racial hatred, xenophobia, anti-Semitism or other forms of hatred based on intolerance, including: intolerance expressed by aggressive nationalism and ethnocentrism, discrimination and hostility against minorities, migrants and people of immigrant origin.

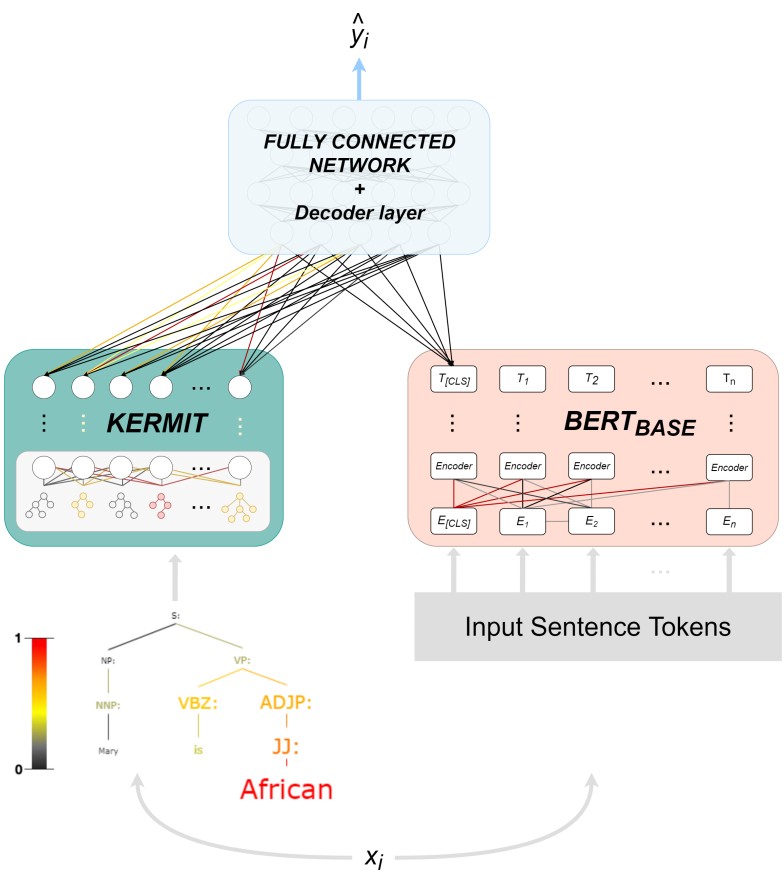

**Figure 2** **KERM-HATE architecture, forward and interpretation pass.**

[1]The code to generate our model is available at www.github.com/ART-Group-it/HateSpeechKermit.

## An Explainable Syntax-based Hate Speech Recognizer

Our Hate Speech Recognizer stems from a recent result on visualizations of activations of syntactic trees for decisions of neural networks (*Zanzotto et al., 2020*).

We then propose an Unintended-bias Visualizer based on Kermit modeling (KERM-HATE[1]) that is a model for hate speech recognizer consisting basically of three components:

1. KERMIT model (*Zanzotto et al., 2020*).
2. A transformer model.
3. A fully-connected network.

The structure of KERM-HATE (see Fig. 2) makes it a particular model, because it combines the syntax offered by KERMIT with the flexibility of transformers and the possibility to switch a representation space $\mathbb{R}^n \rightarrow \mathbb{R}^m$ of a fully connected network.

The first component, that is, KERMIT, allows the encoding and the visualization of the activations of universal syntactic interpretations in a neural network architecture.

KERMIT component is itself composed of two parts:

1. KERMIT encoder, which converts parse tree $T$ into embedding vectors, and a multi-layer perceptron that exploits these embedding vectors.

2. KERMIT*viz*, its viewer, makes KERMIT the most relevant component inside KERM-HATE.

KERMIT*viz* gives the possibility to extract as output not only the classification target but especially the colored parse tree with the activation value of every single node that composes a generic sentence (see Fig. 1 and Fig. 2). KERMIT*viz* is the real game changer for our purposes as it allows us to visualize how decisions are made according to activations of syntactic structures.

The transformer component consists of BERT (*Devlin et al., 2018*). KERM-HATE uses the base version of BERT with uncased setting and pre-trained in English language.

The final component is a four-layer fully-connected neural network and boost performances. The component acts on the concatenation of KERMIT output and BERT output. In this network, KERM-HATE changes the input representation space four times: $\mathbb{R}^n \to \mathbb{R}^m \to \mathbb{R}^n \to \mathbb{R}^m$. The result is passed to the decoder layer returning the target value.

## Prejudice in machine learning models

Since we rely on explainable machine learning models, we can revitalize the operational definition of prejudice of statistical functions (*Kamishima et al., 2012*) within the context of neural networks:

**Prejudice-in-NN** A learned model shows *prejudice* when takes decisions by overusing identity terms.

This definition is in line with the definition of prejudice given in *Quasthoff (1989)* but differs from the definition of *un-intended bias* given in *Dixon et al. (2018)* that focuses on performances of learned models. Moreover, using this definition, we can provide a measure of degree of perceived prejudice since links prejudice to single decisions of learned models.

## Possibly unbiased training and validation corpus

*Davidson et al. (2017)* collected and annotated a possibly unbiased dataset containing about 25,000 tweets, which can be used for our purposes. In fact, the procedure for collecting and annotating the corpus may have reduced the lexical probabilistic bias, although this procedure has been designed for a different purpose.

The procedure for collecting the Davidson et al. dataset stems from an important point: tweets are selected starting from 1,000 terms from HateBase (www.hatebase.org). Therefore, these tweets contain possibly ethically-charged hate terms. Then, at the end of the annotation, tweets of all classes may contain hate terms. This should guarantee that probabilistic lexical bias is reduced to a minimal amount.

The Davidson et al. dataset was manually annotated using CrowdFlower (CF) workers. Workers were asked to label each tweet as one of three categories: *Hate speech*, *Offensive language* but not hate speech, or *Neither*: non-offensive and non-hate. The distinction between *Hate speech* and *Offensive language* is the real reason behind this corpus. Each tweet was annotated by at least three annotators. The inter-annotator-agreement score provided by CF-workers was 92%. As another factor to reduce ethically charged bias, annotators paid attention to inter-words and statements often labeled as hate speech which are not: an example is *ni\*\*a* labeled as *Hate speech* but used as an interchange

by African-Americans (*Warner & Hirschberg, 2012*). The resulting dataset contains 1,430 tweets labeled as *Hate speech*, 19,190 tweets as *Offensive language* and 4,163 tweets as *Neither* (non-offensive and non-hate).

Then, Davidson et al. dataset can be used for our study as it should help in disentangling the positive or negative use of some given hate words.

## Challenging hate speech recognizers on hot topics

This is another important aspect of our study. We aimed to collect fresh corpora generated by identifiable groups of people and which may contain potentially offensive or hate speech utterances. After an initial analysis for selecting the topics ('Two discussed and divisive topics of 2020' section), 'Black Lives Matter corpus' and 'United States presidential election corpus' sections describe how we collected the two novel corpora.

### Two discussed and divisive topics of 2020

The year 2020 has been the year of the pandemic. Nevertheless or, possibly, exacerbated by that, 2020 has had some triggering events generating real conflicting situations, which reverberated in the social media arena. Some of these events, unfortunately, have touched open wounds, which are dividing people of United States in different, well-identified social groups.

Two events have the right characteristics for our study:
1. *The Black Lives Matter protest* –started on May 25, 2020 in Minneapolis (Minnesota) triggered by the death of George Floyd.
2. *The 2020 American presidential election* held on November 3, 2020.

In the Black Lives Matter protest the black community and also other minorities fight against social injustice. This tremendously divisive event generated a great production of tweets originated in the black community, whose language is generally labeled by HSRs as offensive (*Williams & Domoszlai, 2013*; *Anderson et al., 2018*). On the other hand, the presidential election sees a strong conflict of political ideologies between the two major parties in the US: the Democratic and Republican parties. Here, there is not a social group whose language is generally labeled as offensive by HSRs. Yet, given the electoral campaigns of the two parties, the tweet corpus can be easily split in two parts related to two different groups of American voters.

### Black Lives Matter corpus

For gathering possibly non-offensive utterances produced during the Black Lives Matter (BLM) protest and targeting the black community, we used a proxy event: the release of the movie *Black is king*. This movie emphasizes gender and race equality and should have induced pride in the black community. The movie's messages are included in those of the BLM protest, which makes it a possible propaganda medium (*Crumpton, 2020*; *Woronzoff, 2020*). Using these messages, we can create a corpus containing user opinions on a clear, direct and pop topic that is based on messages of inclusivity and peace. The idea, is to get a corpus with as many ideologies as possible - even in disagreement with each other - on a topic that touches on sensitive issues but only with positive implications.

**Table 1  Black Lives Matter Corpus example sentences.**

| # | Sentence |
|---|----------|
| 1 | If I don't like this movie does it make me racist ? |
| 2 | I'm proud to be an African. |
| 3 | Black people are the worst to each other. |
| 4 | All white people are racist. |
| 5 | The way you represent blacks is just second to none, keep on |
| 6 | Sounds like racial superiority. nazis thought just like that. |
| 7 | Black supremacist language trending. |
| 8 | White fragility is at an all time high. |
| 9 | I get more hype every day waiting for black is king. |

The corpus has been collected in two different time slots—on July 20, 2020 and on August 1, 2020—which are one day after two triggering events, respectively, the disclosure of the trailer and the release of the movie. To augment variety, we performed the first extraction on Twitter and the second on Instagram.

In the first time slot, we collected 7k tweets geolocated in the US containing the hashtag `#BlackIsKing` using Tweepy. Then, we extracted only one tweet for each account, obtaining a sample of 658 tweets. Selecting one tweet for account augment diversity.

In the second time slot, we extracted comments from 7 Instagram public posts about the movie: posters, previews and emerged discussions about the actors who starred in it. Selected posts were public posts and, therefore, it is possible to view them even without having an Instagram profile. Usernames that appeared in a comment were deleted for privacy. Our bot[2], based on ChromeDriver, collected 2,650 comments. So, our final *Black Lives Matter corpus* consists of 3,308 comments and tweets.

In Table 1 we show some sentences found in the BLM corpus. We reported a sample of sentences in which at least one potentially offensive keyword is included.

### United States presidential election corpus

The second analyzed event of 2020 was the United States presidential election held on November 3, 2020. Our aim was to generate a corpus based on two datasets containing tweets of Americans politically aligned with the two major contemporary political parties in the United States: the *Democratic party*—with Joe Biden as candidate—and the *Republican party*—with Donald J. Trump as candidate. In Kaggle there are two datasets about the US 2020 election[3]: the first dataset—with 775,054 tweets—contains all the tweets having `#Biden` or `#JoeBiden` as hashtags, while the second contains 958,580 tweets having `#Trump` or `#DonaldTrump` as hashtags. Both datasets were generated from October 15, 2020 to November 8, 2020 and hold for each tweet 11 other fields including geolocation. Using this information, we filtered all tweets written in the United States obtaining only tweets

[2]Available at www.github.com/itsmattei/ Catch-Instagram-post-comments.

[3]Datasets available at www.kaggle.com/ manchunhui/us-election-2020-tweets.

**Table 2** Twitter verified profiles selected for each party.

| Name | Twitter username |
|------|------------------|
| Joe Biden | `@JoeBiden` |
| Kamala Harris | `@KamalaHarris` |
| Democratic Party | `@TheDemocrats` |
| Donald J. Trump | `@realDonaldTrump` |
| Mike Pence | `@Mike_Pence` |
| Republican Party | `@GOP` |

☐ Democratic party    ☐ Republican party

created by possible American voters. Although the two datasets contain the names of presidential candidates this does not imply that they are politically aligned.

Accordingly, we studied the hashtags and slogans used by the major exponents of the two political parties. For each party, we selected the verified Twitter account of: the presidential candidate, the vice president candidate and their political party. Table 2 shows the Twitter profiles analyzed for each party.

Then, in each group, we compiled a ranking list of the most used hashtags (the list of hashtags most used by both parties is shown in Appendix A). Finally, we generated our political datasets filtering from the Kaggle dataset holding only the tweets geolocated in the US, all the tweets with at least one of the hashtags present in the lists. In particular, regardless of whether the dataset contains the hashtag #Biden rather than #Trump, a tweet is considered democratic—and therefore included in the appropriate dataset—if it has at least one hashtag present in the democratic list. The same is applied to the Republican dataset. So we obtained a *Democratic dataset* of 46,898 tweets and a *Republican dataset* of 35,903 tweets. So, our *United States presidential election corpus* is composed of 82,801 tweets.

In Table 3 we show some sentences in both the *Democratic dataset* and *Republican dataset* that make up the *United States presidential election corpus*. Unlike the BLM corpus (ref. 'Black Lives Matter corpus' section) - where the potentially offensive keywords are more obvious (ref. Table 1), in this case, we included sentences that clearly and concisely explained the political orientation of voters.

To test whether the two datasets that make up the corpus actually reflect the political ideologies of American voters, we analyzed the geolocation of tweets and compared them with the National Exit Polls. In all states where the number of geolocated tweets was relevant, we can confirm how the prevalence of tweets placed in a specific dataset actually reflected the outcome of the exit poll. A detailed list of the states that most affected the two datasets—and those that have a larger gap—is shown in Appendix B.

**Table 3  United States presidential election corpus example sentences.**

| # | Sentence |
|---|---|
| 1 | He proves that he is the worst president EVER. |
| 2 | Good morning Twitter democratic voters. |
| 3 | As a supporter you should probably avoid words like 'immoral'. |
| 4 | Damn right I support him 100 percent. |
| 5 | Biden has always worked to help stuttering kids. |
| 6 | Democrats think the Constitution is more important than President. |
| 7 | The Evilness of human beings should be measured in TRUMPS. |
| 8 | Glad u got out of the house! D**K! |
| 9 | Donald Trump 4 more years! |

## EXPERIMENTS

Our experiments are designed to investigate two different issues:

1. Assessing if syntactic information of sentences is useful for defining models for hate speech recognition with higher performances.
2. Determining if syntactic information has the power to wipe out prejudice in learned models.

The rest of the section is organized around the two above issues. Firstly, 'Experimental set-up' section gives the general settings of our experiments. Then, 'Results and discussion' section reports on results on the experiments for the two different issues adding additional settings when necessary.

### Experimental set-up

For the first set of experiments, which is devoted to understand whether explicit syntactic representation can be useful in building hate speech recognizers (HSRs), we experimented with our model KERM-HATE over three different already annotated datasets: the Davidson et al., the Waseem & Hovy and the Founta et al. dataset. The Davidson et al. dataset is our main annotated dataset and it is used also to train the final model (see 'Possibly unbiased training and validation corpus' section). The Waseem & Hovy dataset consists of 16,849 tweets annotated in three classes (*Racism*, *Sexism* and *Neither* (non-racism and non-sexism). Finally, the Founta et al. dataset consists of 91,951 tweets annotated in four classes (*Abusive*, *Hateful*, *Normal* and *Spam*). For the experiments, these datasets have been randomly split in 80% for training and 20% for testing.

In this first set of experiments, we compared our approach to three HSRs based syntactic-agnostic transformers, that is, $BERT_{BASE}$ (*Devlin et al., 2018*), *RoBERTa* (*Liu et al., 2019*) and *XLNet* (*Yang et al., 2019*), and to two basic explicit-syntactic-aware models, that is KERMIT *BERT* and KERMIT$_{XLNet}$ (*Zanzotto et al., 2020*). Finally, as suggested in *Zanzotto et al. (2020)*, we explored also two special versions of BERT, that is, $BERT_{REVERSE}$ and $BERT_{RANDOM}$, which constitute the core for testing if the task is syntactically-sensitive.

In detail, let $S = \{w_1, \ldots, w_n\}$ a sentence consisting of $n$ words ($|S| = n$). In $\text{BERT}_{REVERSE}$, the input sentence $S$ is given in reverse way $S = \{w_n, w_{n-1}, w_{n-2}, \ldots, w_1\}$ while in $\text{BERT}_{RANDOM}$ the input sentence $S$ is given in random way $S = \{w_i, w_j, w_k, \ldots, w_z\}$ (with $i \neq j \neq k \neq z$ and $i, j, k, z \leq n$). To assess statistical significance, each experiment is repeated 10 times with different seeds for initial weights.

The meta-parameters utilized in training the models are the following and so for KERM-HATE, KERMIT *BERT* and KERMIT $_{XLNet}$:

1. The tree encoder is on a distributed representation space $\mathbb{R}^d$ with $d = 4000$ and has penalizing factor $\lambda = 0.4$ (as suggested for tree kernels in *Moschitti (2006)*).

2. Constituency parse trees have been obtained by using Stanford's CoreNLP probabilistic context-free grammar parser (*Manning et al., 2014*).

KERM-HATE's fully-connected four-layers network change the representation space four times: $\mathbb{R}^n \rightarrow \mathbb{R}^m \rightarrow \mathbb{R}^n \rightarrow \mathbb{R}^m$ where $m = 2,000$ and $n = 4,000$, before concluding with the final classification layer. A dropout layer (*Srivastava et al., 2014*) with 0.1 is added for each layer. Class weight $w_i$ is inversely proportional to its *class$_i$* ($C_i$) cardinality ($w_i = \frac{1}{|C_i|}$). $\text{BERT}_{BASE}$, RoBERTa and XLNet, both stand alone or as components of KERM-HATE, was implemented using Huggingface's transformers library (*Wolf et al., 2019*). The input text for $\text{BERT}_{BASE}$ has been preprocessed and tokenized as detailed in *Devlin et al. (2018)*. The optimizer used to train all the models is AdamW (*Loshchilov & Hutter, 2019*) with the learning rate set to $2e^{-5}$. All models used a batch size of 32 and are trained for 4 epochs. Our hardware system consists of: 4 Cores Intel Xeon E3-1230 CPU with 62 Gb of RAM and 1 Nvidia 1070 GPU with 8Gb of onboard memory.

The second set of experiments, which aims to understand if syntax can help in wiping out prejudice, is organized on three tests –the *Blind test*, the *Inside out test* and the *Prejudice test* –carried on our novel collected corpora (described in 'Black Lives Matter corpus' and 'United States presidential election corpus' sections) with the help of 24 different annotators. The partition of annotators into the three tests does not depend in any way on their cultural background since it is for all different with greater or lesser knowledge of syntax, machine learning and natural language processing. In addition, none of the annotators was born or stayed - during the execution of the test - in the United States of America and nobody has any emotional ties with that land. Finally, all the annotators do not belong to the ethnic groups listed and used in the tests. These restrictions have been used in order to make tests as objective and clear as possible. A methodological note: the design of these three tests is sequential, that is, the result of one test led to the definition of next one. Hence, questions to annotators posed in *test$_i$* depend on the results obtained from *test$_{i-1}$*. Limitations of *test$_{i-1}$* lead to the the selection of sentences presented in *test$_i$*. Fleiss'kappa score (*Fleiss, 1971*) measures the inter-annotator agreement. In all these tests, we used $\text{BERT}_{BASE}$ and KERM-HATE trained on the Davidson et al. dataset.

The *Blind test* aims to check the accuracy of $\text{BERT}_{BASE}$ and KERM-HATE on a novel, divisive datasets. This test is a classical annotation with respect to the guidelines used in the Davidson et al. dataset. The question posed to annotators for each sentence was: "*Based on the guidelines you read, how would you label this sentence?*". The possible answers were the targets given by the Davidson: *Hate speech*, *Offensive language* and *Neither*. Annotators are

**Table 4** Performance of the Hate speech recognizers on the three different datasets. Mean and standard deviation results are obtained from 10 runs. The symbols ◇, † and ∗ indicate a statistically significant difference between two results with a 95% of confidence level with the sign test.

| Model | Davidson dataset | | Waseem and Hovy dataset | | Founta dataset | |
|---|---|---|---|---|---|---|
| | Average Accuracy | Average Macro F1 score | Average Accuracy | Average Macro F1 score | Average Accuracy | Average Macro F1 score |
| $BERT_{BASE}$ | 0.67 ($\pm$ 0.03)◇ | **0.48 ($\pm$ 0.02)**◇ | 0.73 ($\pm$ 0.01)◇ | **0.50 ($\pm$ 0.09)**◇ | 0.54 ($\pm$0.02)◇ | **0.46 ($\pm$0.01)**◇ |
| $BERT_{REVERSE}$ | 0.66 ($\pm$ 0.01) | 0.47 ($\pm$ 0.01) | 0.54 ($\pm$ 0.12) | 0.34 ($\pm$ 0.07) | 0.49 ($\pm$0.07) | 0.38 ($\pm$0.04) |
| $BERT_{RANDOM}$ | 0.66 ($\pm$ 0.02) | 0.47 ($\pm$ 0.01) | 0.50 ($\pm$ 0.10) | 0.33 ($\pm$ 0.07) | 0.47 ($\pm$ 0.08) | 0.38 ($\pm$ 0.04) |
| XLNet | 0.47 ($\pm$ 0.06)† | 0.34 ($\pm$ 0.03)† | 0.55 ($\pm$ 0.08)† | 0.39 ($\pm$ 0.10)† | 0.53 ($\pm$0.03)† | 0.42 ($\pm$0.01)† |
| RoBERTa | **0.78 ($\pm$ 0.01)** | 0.37 ($\pm$ 0.05) | **0.73 ($\pm$ 0.12)** | 0.44($\pm$ 0.09) | **0.59 ($\pm$0.05)** | 0.42($\pm$0.03) |
| $KERMIT_{BERT}$ | 0.72 ($\pm$ 0.02)∗ | 0.54 ($\pm$ 0.02)∗ | 0.79 ($\pm$ 0.01)∗ | 0.54 ($\pm$ 0.09)∗ | 0.60 ($\pm$ 0.02)∗ | 0.51 ($\pm$ 0.01)∗ |
| $KERMIT_{XLNet}$ | 0.68 ($\pm$ 0.05)† | 0.47 ($\pm$ 0.03)† | 0.74 ($\pm$ 0.02)† | 0.51 ($\pm$ 0.10)† | 0.56 ($\pm$ 0.03)† | 0.47 ($\pm$ 0.01)† |
| **KERM-HATE** | **0.80 ($\pm$ 0.02)◇∗** | **0.66 ($\pm$ 0.01)◇∗** | **0.91 ($\pm$ 0.02)◇∗** | **0.86 ($\pm$ 0.04)◇∗** | **0.64 ($\pm$ 0.02)◇∗** | **0.54 ($\pm$ 0.01)◇∗** |

unaware of the choice generated by the two models. For this reason, the test is called *blind*. In this test, we randomly selected 34 sentences from the two corpora and each sentence received 10 annotations from the 10 different annotators. The final label given to the sentence is assigned with a majority vote.

The *inside-out test* aims to evaluate if the annotators agreed or disagreed with the labeling given by KERM-HATE model given the post-hoc explanation of the *heat parse tree*. This test contains 25 sentences with an overlap of 30% of the sentences in the *Blind test*. 10 annotators participated in this test. The question posed was: "*Based on the label and parse tree activation values, do you agree with the label given by KERM-HATE?*" The possible answers were *Yes* or *No*. Moreover, if the answer was *No*, it was asked to indicate what the reasons were. The name of the test comes exactly from the possibility given to the annotators to look *inside the model* in order to understand its output.

The *Prejudice test* aims to understand whether classification of KERM-HATE model relies upon prejudice: ethnicity, geographic or gender bias. The set of sentences is the same as *Inside out test*. Sentences are given along with the classification of KERM-HATE and the *heat parse tree* The question posed was: "*Based on the label and parse tree activation value, do you think this label was obtained through any bias?*". The possible answers were *Yes* or *No*. Also in this case—if the answer was *Yes*—it was asked to specify the reason why the system decision is biased. 4 annotators participated in this test and each annotator answered questions for the 25 sentences.

## Results and discussion

Syntactic information is useful to significantly increase performances of Hate Speech Recognizers (HSRs) (see Table 4) and KERM-HATE is the best model. The preliminary test of syntactic-sensitive task seems to suggest that the hate speech phenomena is quite sensitive to syntax. Indeed, in Waseem & Hovy and Founta et al. datasets the performance of $BERT_{BASE}$ is significantly better than $BERT_{REVERSE}$ and $BERT_{RANDOM}$. However, in Davidson et al. dataset the gap between the three models is smaller but still in $BERT_{BASE}$ favor in all tests conducted. Nevertheless, in the three annotated datasets, KERMIT and $KERMIT_{XLNet}$ significantly outperform $BERT_{BASE}$ and XLNet, respectively. The first pairs

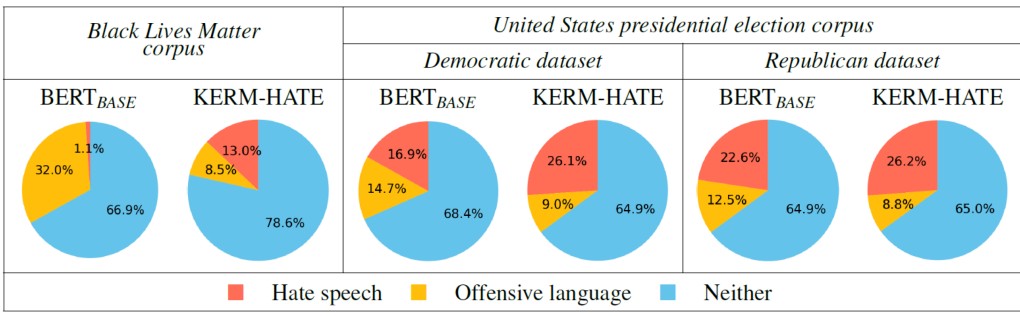

**Figure 3** Labeling phase on our generated corpora using BERT<sub>BASE</sub> and KERM-HATE.

**Table 5  Test summary.** For each test, inter-annotator agreement and the results obtained are reported.

| Test | Blind | | Inside-out | Prejudice |
|---|---|---|---|---|
| Fleiss' Kappa | 0.19 | | 0.24 | 0.87 |
| | Accuracy | | Average Agreement | Average Perceived |
| | BERT<sub>BASE</sub> | KERM-HATE | with KERM-HATE | Prejudice |
| | 0.11 | 0.41 | 0.52(±0.08) | 0.55(±0.04) |

of models use explicit syntactic information whereas the second pair does not. Moreover, our KERM-HATE outperforms all the other models, including RoBERTa, which has a very high average accuracy.

Hence, KERM-HATE trained on Davidson et al. (see 'Possibly unbiased training and validation corpus' section) is a good candidate for exploring whether or not it holds the prejudice of the corpus where it is trained.

On the novel, divisive corpora (ref. to 'Two discussed and divisive topics of 2020' section), trained KERM-HATE and BERT<sub>BASE</sub> behave differently (see Fig. 3). Then, these two models definitely look at different features of input sentences. Generally, KERM-HATE seems to be less prone than BERT<sub>BASE</sub> to tag sentences as *Offensive language*. On the other hand, BERT<sub>BASE</sub> is oriented to tag sentences as *Offensive language* in the *Black-lives-matter corpus* and this predisposition can also be seen in the other two datasets.

The *Blind test* on the novel corpora confirms that KERM-HATE is better than BERT<sub>BASE</sub> (see Table 5). In fact, BERT<sub>BASE</sub> matches only 11% of the labels assigned with majority vote by annotators whereas KERM-HATE matches 41% of the labels. However, hate speech recognition in these corpora is rather difficult. In fact, the inter-annotator agreement on the blind test is 0.19 (*Slight agreement*).

Moreover, decisions of KERM-HATE seems to convince annotators when presented along with explanations. In the *Inside-out test*, annotators have an average agreement with KERM-HATE of 0.52(±0.08), which is higher with respect to the *Blind test* (see Table 5). The task confirms to be a very subjective task (*Basile et al., 2021*) as the inter-annotator agreement is 0.24 (*Fair agreement*). According to the analysis of this group of annotators, when KERM-HATE fails, the reason seems to be that it focuses on words correlated with

[4]The complete list is in the Appendix C in the additional material.

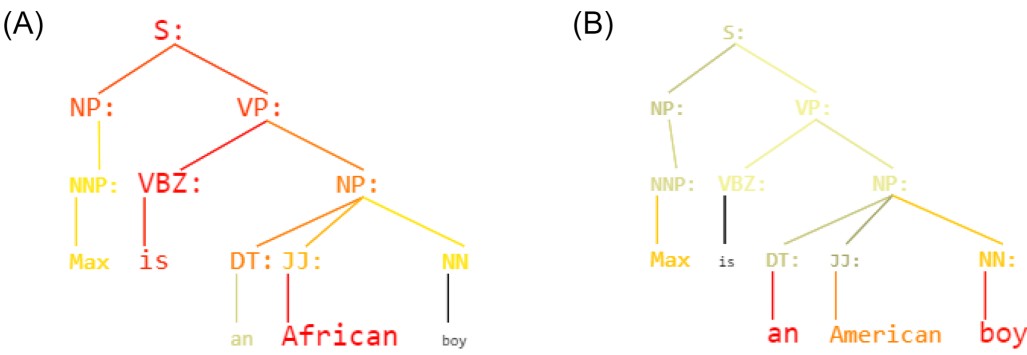

**Figure 4** (A) Sentence: *Max is an African boy* - Labeled as: Offensive language (B) Sentence: *Max is an American boy* - Labeled as: Neither. KERM-HATE colored parse trees output.

ethnicity, continents or nations, proper names and gender[4]. This test suggests that the system is still ethically-biased and, thus, we performed the *Prejudice test*.

In contrast with what hypothesized, syntactic information does not wipe out prejudice from our HSR. In fact (see Table 5), the average perceived prejudice of KERM-HATE is definitely high, that is, 0.55(±0.04) and annotators have the very high agreement of 0.87 on this fact (*Almost perfect agreement*).

## Sample perturbation analysis

Since KERM-HATE provides heat parse trees as a post-hoc explanation of decisions, it offers the opportunity to carry out perturbation analysis.

The *perturbation analysis* is a handy method to analyze if comments are really classified with prejudice. The basic idea is to replace one or more keywords in a sentence $S_A$ generating a sentence $S_B$. The action carried by $S_A$ is the same as $S_B$ but participants or their adjectives are altered. With this operation, the colored syntactic trees derived from KERM-HATE of $S_A$ and $S_B$ should be different and it will be possible to compare them and visualize eventual syntactic biases.

In the Fig. 4 is shown an example: given the sentence *"Max is a [W] boy"*, KERM-HATE reacts and uses differently parse trees if W is *"African"* (A)) or *"American"* (Figure 4B). Firstly, the classification is different: *'Offensive language'* and *'Neither'*, respectively. Secondly, the active part of the parse tree is different. For the sentence with *"African"*, the structure [Subject] is [Object: African] seems to be particularly active. Conversely, for the sentence with *"American"*, this part loses importance that is gained by the second noun phrase. This shows how decisions are correlated with divisive terms. Other examples are given in the additional material (Appendix D).

## CONCLUSIONS

Hate speech recognizers (HSRs) are a dual-use technology that may help in controlling spread of online hate but these can be used as a way to impose a, possibly unintended, censorship. In these paper, we show that even apparently better models may hide a very high

level of prejudice captured from training corpora. Actually, KERM-HATE, our explainable syntactic HSR, has unveiled this fact.

Hence, our study suggests that HSRs are still a technology prone to prejudice and should be handled with care. Nevertheless, our explainable syntactic HSR has opened the route to spot why HSRs have prejudice and, possibly, finding recovery strategies by defining *ad-hoc* rules for mitigating this unintended prejudice.

### Funding
This research was funded by the 2019 BRIC INAIL ID32 SfidaNow project. There was no additional external funding received for this study. The funders had no role in study design, data collection and analysis, decision to publish, or preparation of the manuscript.

### Grant Disclosures
The following grant information was disclosed by the authors:
The 2019 BRIC INAIL ID32 SfidaNow project.

### Competing Interests
The authors declare there are no competing interests.

### Author Contributions
- Michele Mastromattei conceived and designed the experiments, performed the experiments, analyzed the data, performed the computation work, prepared figures and/or tables, authored or reviewed drafts of the paper, and approved the final draft.
- Leonardo Ranaldi performed the computation work, authored or reviewed drafts of the paper, and approved the final draft.
- Francesca Fallucchi conceived and designed the experiments, analyzed the data, prepared figures and/or tables, authored or reviewed drafts of the paper, and approved the final draft.
- Fabio Massimo Zanzotto conceived and designed the experiments, performed the experiments, analyzed the data, authored or reviewed drafts of the paper, and approved the final draft.

### Data Availability
The model code is available at GitHub: https://github.com/ART-Group-it/HateSpeechKermit

The data scraper is available at GitHub: https://github.com/itsmattei/Catch-Instagram-post-comments

The data is available at Zenodo: Michele Mastromattei, & LeonardRanaldi. (2022). ART-Group-it/HateSpeechKermit: Version 1.2 (v.1.2). Zenodo. https://doi.org/10.5281/zenodo.5858346

## Supplemental Information

Supplemental information for this article can be found online at http://dx.doi.org/10.7717/peerj-cs.859#supplemental-information.

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
