# Peer review of "Syntax and prejudice: ethically-charged biases of a syntax-based hate speech recognizer unveiled"

_PeerJ Computer Science, doi:10.7717/peerj-cs.859_

## Round 0.1 · original submission · Major Revisions

· Academic Editor

Major Revisions

We have received four detailed reports about the manuscript. Please prepare your revision and a point by point response letter. Please note we do not expect you to cite references recommended by reviewers unless you feel they are relevant. It will not change the editorial decision if you choose not to include them.

Reviewer 1 ·

Basic reporting

In investigating if and how syntactic information can be used to de-bias hate speech recognizers, this paper presents negative results: even when just syntactic features are considered, prejudice is captured from training corpora and make the application to new datasets not very fruitful.
The advantage of Kermit, the model proposed by the authors, is that it enables a visual, post hoc analysis of the results by exploring the syntactic information used in the decision process of a neural network. Consequently, it is possible as future work to define ad-hoc rules for mitigating the bias, even if the authors do not clarify how this feedback can be incorporated in the model.

The paper is sometimes not very clear or incomplete. In my opinion, the authors should provide

- a short definition for hate speech recognizers in the introduction
- in the related work section, they should quote works about gender-based bias in neural models

A native speaker should check the paper; here is a list of several mistakes:

Line 41- 42 There is a repetition "aiming in transformers aiming to put trigger words in contexts"
Line 90 - 91 "learned models fail to capture another important fact: learned models can "right from the wrong reasons" -> "learned models fail to capture another important fact: learned models can be "right from the wrong reasons"
Line 100 - 101 "help in shedding the light" -> "help in shedding light"
Line 113 - 114 "Zanzotto et al. (2020)" is not reported correctly (add a verb)
Line 130 "Devlin et al. (2018)" is not reported in the correct way
Could you standardize the name using a letter for the model Kermit with a thunder symbol?

Experimental design

line 35-36 The authors wrote "Ethically-charged biases are extremely more dangerous than simple probabilistic biases, which can lead to a conclusion utilizing wrong premises." However, I think that even the first type of bias is probabilistic in nature, in the sense that the model created on training data acquires their bias probabilistically. In other words, I don't see this dichotomy between probabilistic vs. ethical bias since even ethical bias is acquired through a probabilistic process. Statistical bias becomes ethical when we recognize them as such.

Kermit, a syntax-based hate speech recognizer, is trained on a possibly unbiased training set and tested on two newly created datasets. The description of the datasets is not clear, and sometimes they are reported as corpus. I do not think that the way the new datasets have been created made them meaningful for the reported experiments. For example, the authors suppose that Black lives matter corpus is composed by tweets and comments from Instagram commenting on a movie produced by users from the black community. However, comments and tweets about the movie can be produced by users from different ethnicity. Since the authors can not be sure that the majority of the content came from the black community, the experiment is flawed.

Validity of the findings

Since the datasets created on purpose to investigate the research questions are not well designed, it is not clear what the experiments are showing.

Reviewer 2 ·

Basic reporting

- The structure of the paper is clear and reasonable.

- The form of the sentences needs to be corrected, the beginning of a sentence must be capitalized and the end of a sentence must have a period, e.g, lines: 52, 53, 106-111, ...

- I don't understand the "SaettaKermit" concept because it only appears in Table 2.

- Sentences and grammar need to be sent to proof editing for correction to improve.

Experimental design

- You need to describe more clearly the process of creating the dataset, how to annotate it, how to calculate inter-annotator agreement? It is necessary to analyze and compare datasets on many different linguistic aspects. For example, this paper (https://arxiv.org/pdf/2103.11528.pdf) describes how to create a dataset for HSD.

- The evaluation indicators have not been clearly described.

- There should be an ablation test to clarify the role of syntax integration.

- The perturbation contribution needs to be further analyzed with clearer data. I don't know what the number is and how much your method solved.

Validity of the findings

Integrating syntactic information into models is essential. This paper has proved its effectiveness on the HSD task. But you need to clarify my suggestions to increase the persuasion.

Reviewer 3 ·

Basic reporting

- p.1 : what the authors mean with the phrase "ethically charged" can be misleading. The expression in itself can make one think of something that bear a positive connotation (i.e. something that is supposed to do some good and has a positive societal impact), but from what is stated in the immediately following sentences, it would seem to indicate the exact opposite, that is a type of bias that brings potential representational harms to certain groups

- nodes in the parse trees should be made more visible; I understand that the purpose is to provide a visual representation of the different activation values, by foregrounding the active nodes, but in some figures the other nodes (especially the black ones) are barely readable, while I think it's important to give a clear picture of the whole sentence, also showing the nodes that remained inactive (the color scale used by the authors is indeed helpful in this)

- p.3, ll. 82-3: is it "unintended" bias, instead of "untended"? I don't seem to find the reported quotation in Hardt et al. 2016

- p.8. ll.304-5: I don't fully understand what the authors mean when they say that Davidson et al.'s dataset is not sensitive to syntax; they motivate this claim commenting on the same accuracy reported for the three BERT-based models, but they should probably try to elaborate more on this, maybe explaining in particular what BERTreverse and BERTrandom models are useful for (besides just pointing to Zanzotto et al. 2020).
- ll. 310-1: it is not clear why the dataset of Davidson et al. should be a good candidate for experiments related to the second research question, while it is not the case for that of Waseem and Hovy, which is the one on which SaettaKermit gave the best results ever. Wouldn't it be worth replicating the experiments of round 2 on this dataset as well?
- ll. 314-15: the data shown in the figure seem to contradict what is stated in this sentence: from what I see in the other two datasets BERTbase also prefers, in non-"neutral" tweets, the classification of the tweet content as hateful, rather than as offensive (although in the Democrats dataset this difference is much less marked).

- Appendix C: 1) it should be made clearer in the text that sub-figures b in the examples are those were one or more words were changed
2) line 472 says "parse trees obtained from KERMITbert", but based on what described in sect. 4.3 the qualitative analysis was carried on the parse trees from SaettaKermit, so using examples from a different model sounds very confusing. Should I assume this is just a typo or is there any other reason?

Typos:
- l. 75: dialectic -> dialect (?)
- non-syntactic citations should be put in brackets, e.g. in line 114: " [...] for decisions in neural networks (Zanzotto et al. 2020). "
- l.91: models can "right for the wrong reasons" -> can be right (?)
- l.116: model for hate speech recognizer -> recognition
- l. 162: it should helps -> help
- l. 319: has matches-> matches (?)
- Appedix C: the parse tree i Fig. 6a is exactly the same as the one in Figure 6b

Experimental design

- p.6+Appendix A: (on the creation of the US elections dataset) an hashtag is not necessarily an endorsement, so it is not clearbased on what the presence of a hashtag referring to Biden or Trump can establish in such a deterministic way the political orientation of the user. Further details are provided in Appendix A about this, but it is not clear what the percentages in Table 5 actually refer to; is it the proportion of tweets in the respective dataset? Furthermore, the authors try to show that the choice of using the hashtag as a proxy for the assignment to one of the two datasets is reflected in the data on exit polls. However, the data given in the dedicated table 1) are not systematic (for some states the proportions are reversed), 2) the numerical difference does not appear so striking, except in a few cases 3) the table is incomplete, in that the data is not shown for all states, but only for those in which - according to the authors - the gap is greater. If, as it seems to understand, the gap refers precisely to the percentage of votes given to one of the two parties, the use of the hashtag as a discriminating factor seems to be a fairly weak motivation

- p.7: the sample used for the Blind test and follow-ups is quite small, especially considering the original size of the novel datasets created for the experiments. Is there a specific reason for such a choice? Also, what background did the annotators have? Were they under/graduate students, crowdworkers, experts on syntax/deep learning, other? This seems to be a relevant information to add, especially to help the reader understand the possible reasons behind the low IAA results

Appendix C: Fig. 5a: I don't see how this can be considered as an instance of offensive tweet. Is the example taken from the sample of tweets used for the second round of experiments? What I mean is whether human annotators actually agreed on the final label provided by Kermit

Validity of the findings

no comment

Additional comments

The paper introduces KERMITz (or SaettaKermit, as I will call it henceforth in the detailed comments), a visual tool that extends a syntax-based hate speech recognizer (Kermit), and describes the experiments carried out with such tool. More specifically, the two research questions the authors aimed to explore in this work are 1) whether the use of syntactic information is helpful and contributes to the improvement of the performance of a hate speech classifier, 2) if the use of such information can help remove possible biases, which other models, based mainly on lexical information alone, may tend instead to reproduce and amplify. With respect to the first point, the reported experiments show how the use of syntax - in the form of constituent trees - significantly increases performance. As for the second, however, the experiments carried out show that even the use of syntax is not able to reduce the presence of biases in the system's decision-making process.
Precisely the negative results detected with respect to the latter point represent the main contribution of this work, together with the possibility of using a hate speech recognition system that also provides a visualization tool for post-hoc analyses of the system's decisions.
The paper, however, is not always easy to follow, and relevant information is sometimes reported in a confusing or incomplete way.

·

Basic reporting

The article is generally well written and clear, but there are some typos:
Line 141: the definition [...] that focusES (not "focus")
Line 156: language is THE real reason
Line 162: as it should help (not "helps")
Line 187: 3.4.2 Black Lives Matter corpus (with capital "L" and "M")
Note 3, Line 202: Available at (with capital "A")
Line 205: was the United States presidential election (without "about")
Line 315: and this predisposition (without the "-")
Appendix A, Line 458: two datasets; (2) states (put a semicolon between point 1 and 2!)
Appendix B, Line 463: if they agreed (using "he/she" is correct too, but the neutral "they" is also a good choice in standard formal English)

Also, pay attention to bibliographic references, because some of them are not between brackets:
Line 114: for decisions of neural networks Zanzotto et al. (2020) > for decisions of neural networks (Zanzotto et al. 2020).
Line 130: The transformer component consists of BERT Devlin et al. (2018) > The transformer component consists of BERT (Devlin et al. 2018).

Tables and figures are clear, but it is fundamental to correct:
Table 2, bottom line: Is "SaettaKermit" KERMIT(with the bolt in subscript)? You never call it SaettaKermit in the rest of the article, so, please, align "SaettaKermit" with your usual terminology, or at least explain in the notes what "SaettaKermit" is;
Appendix C, Figure 6 (FUNDAMENTAL!): Example (a) has the same image as example (b). Please, change it with the correct one (Mary is American), because Figure 6's examples are really powerful and useful for the paper's comprehension.

The Introduction should contain examples of the prejudice you are describing, especially at Line 33, and at Line 39-40-41. Otherwise, it might appear that you believe that hate speech producers are somehow unfairly censored. For this reason, it might be useful for you to specify that this kind of unjust censorship is applied to those who produce non-hateful messages, thus also involving marginalized people who are simply describing their experiences (such as "Mary is African", from Appendix C).
Also, it might be a good idea to specify that hate speech moderation is not censorship. This is motivated by the fact that hateful messages make their targets less prone to communicate and to express their point of view, thus de facto censoring them. This is the reason why preventing hate speech is a meaning to protect freedom of speech. Useful bibliography on this subject:
West, C. (2012), Words That Silence? Freedom of Expression and Racist Hate Speech, in I. Maitra & M. K. McGowan (eds.) Speech and Harm. Controversies Over Free Speech, Oxford, University Press Scholarship Online.

VERY IMPORTANT: The Introduction should contain a definition of hate speech, as hate speech currently does not have a universally accepted definition.

In 4.1 Experimental Set-up you should include the inter-annotator agreement in the description of the Blind/Inside-out/Prejudice Test's, in each passage.

Experimental design

No comment

Validity of the findings

All underlying data have been provided; they are robust, statistically sound, & controlled: I am uncertain about this.
The methodology used is generally solid, the underlying data have been provided and the experiments can be replicated.
However, in my opinion, I think some sections could be improved.
1) 3.4.2 Black lives matter corpus: I am not convinced by the choice of investigating "non-offensive utterances produced during the Black Lives Matter (BLM) protest and written by the black community" through gathering tweets and Instagram posts about a movie. "Black Is King" is indeed a significative movie for the Black community, but you cannot be sure that only Black people commented it on Twitter/Instagram. Although this corpus is not without potential for studying BLM movement, perhaps a corpus composed of tweets directly about the BLM movement could have been more adequate, albeit also with more noise and hate speech.
2) 4.1 Experimental Set-up (from Line 272): The Blind/Inside-out/Prejudice Test is excellent, but I am not convinced by the number of sentences analysed, especially in the case of the Blind test, from which the sentences analysed in subsequent tests derive. I think 34 sentences are too few, especially if selected randomly from both corpora and if you want to compare the human annotation of HS with the annotation of BERT and KERMIT. There is no need to annotate manually over 6,000 tweets, like in HaSpeeDe 2 task from EVALITA 2020 (Sanguinetti et al., 2020), for example, but at least 100. Also, who are the annotators? You need to include some information about them, such as:
- Are they English native speakers? Or at least they have a high/good comprehension of written English?
- What is their nationality? Are they from the United States, thus emotionally involved in the BLM/presidential election?
- Are there black people among them? There is a study on misogynistic hate speech's recognition that highlights how, in that case, annotators, who are part of the group targeted by the specific hate they are annotating, annotate hate speech differently from the annotators who are not targeted by the same hate speech (Look at Wojatzki et al. (2018) for the full study). Could we be facing a similar case with black people and white people annotating racist hate speech? Even without answering this question, this possibility should be taken into consideration in the paper.
- Do they have any training in annotating hate speech? Have you given them a definition of hate speech?

Sanguinetti M, Comandini G, Nuovo ED, Frenda S, Stranisci M, Bosco C, Caselli T, Patti V, Russo I (2020) Haspeede 2 @ EVALITA2020: overview of the EVALITA 2020 hate speech detection task. In: Basile V, Croce D, Maro MD, Passaro LC (eds) Proceedings of the seventh evaluation campaign of natural language processing and speech tools for Italian. Final Workshop (EVALITA 2020), Online event, December 17th,2020, CEUR Workshop Proceedings, vol 2765. CEUR-WS.org. http://ceur-ws.org/Vol-2765/paper162.pdf

Wojatzki et al. (2018), Do Women Perceive Hate Differently: Examining the Relationship Between Hate Speech, Gender, and Agreement Judgement, in A. Barbaresi, H. Biber, F. Neubarth & R. Osswald (eds.) Pro-ceedings of the 14th conference on Natural Language Processing (KONVENS 2018), pp. 110-120.

Additional comments

This paper discloses important findings for the study and the automatic recognition of hate speech.
It needs to revise its language in a few paragraphs and to improve its data, in order to make them more solid.
I think that the authors have done an excellent job and I believe they should be given the opportunity to revise their text for publication.

---

## Round 0.2 · Major Revisions

· Academic Editor

Major Revisions

A major revision is needed before further consideration. I would encourage the authors to address the issues raised by the reviewers and prepare a detailed response letter.

Reviewer 1 ·

Basic reporting

The paper is significantly improved with respect to the previous version but several sentences are hard to read and there are some typos.

For example:

Line 35 "HSRs aim to recognized posts" -> "HSRs aim to recognize posts". Reformulate the whole sentence
Line 51-54 Reformulate the sentence.
Line 373 "Hate speech recognizers (HSRs) are a dual-use technology as these" -> "Hate speech recognizers (HSRs) are a dual-use technology that"

Experimental design

I think that more details about the datasets created (a couple of concrete examples of the texts that contain) could be useful to better understand the complexity of the task

Validity of the findings

I like the fact that the focus of the paper is on negative results. In this version is more evident.

Reviewer 3 ·

Basic reporting

The paper still needs a thorough proofreading in a number of points.
For example, the paragraph in ll.34-45 is very confusing and not really easy to follow, especially the discussion on censorship and free speech. I see the point, but the very notion of censorship here seems to have first a positive and then a negative connotation. The authors should try to organize the section such that concepts and ideas flow in a more logical way.
In addition, the authors didn't fully addressed reviewers' request to introduce an operational definition of HS. They did, in fact, add references to relevant literature on this matter, but failed to provide a clearer picture of the phenomenon. Highlighting - as the authors rightfully did - the struggles in finding a common definition and the blurry boundaries with related phenomena is perfectly fine, but there actually are some distinguishing traits in HS (e.g. incitement to hate and violence against a specific target), and they should be properly defined in the paper.

- l.107-8: "the above measures of prejudice for learned models fail to capture another important fact:
learned models can be “right from the wrong reasons”": this statement could be further expanded and motivated

Other typos and errors spotted here and there:
- l.89: "inside a neural networks"
- l.92: "Afro American English" -- African-American
- l.99: "untended bias" - - unintended bias (?)
- l.159: "statistical functions Kamishima et al. (2012)" -- (Kamishima et al, 2012)
- ll.243-5: "Finally, we generated our political datasets filtering from the Kaggle dataset holding only the tweets geolocated in the US, all the tweets with at least one of the hashtags present in the lists." -- this sentence should be rephrased
- l.251: "the two datasets made up the corpus actually reflect" -- the two datasets that make up (?) the corpus
- l.293: "proposional to its class" -- proportional
- l.304: "since it is for all different with greater or lesser knowledge of syntax, machine learning [...]" -- please rephrase this sentence fragment
- l.309: "led to the definition of the next." -- next one
- l.310: "questions [...] depends on" -- depend
- Appendix A, l.507: "states those constitute" -- that constitute

Experimental design

- Sect. 3.4.3: The list of hashtags used to filter the tweets should be made available, maybe along with the supplementary material in Appendix A
- Sect. 4.1 and the Blind/Inside-Out/Prejudice tests: the authors in the rebuttal responded to reviewers' comments by motivating the difficutlies in getting a proper pool of annotators, but the main point in those comments didn't have to do with the number of annotators - which is more than reasonable - but with the sample size (i.e. the number of annotated items) used for the analysis, which is very small. With respect to this point, no action has apparently been taken.
Also, for a more systematic description of the annotation process in that step, as well as in the creation of the BLM and US Election datasets, I recommend the authors following the scheme and guidelines proposed here:
https://techpolicylab.uw.edu/wp-content/uploads/2021/11/Data_Statements_Guide_V2.pdf
While this can be done in a concise way, I believe that addressing all the applicable points in these guidelines would help both the authors and the reader in getting a clearer and more exhaustive picture of the data at hand.

Validity of the findings

- Sect. 4.2, l.340: It is not clear what kind of "gap" the authors refer to, when commenting on results in table 2; more in general, the whole sentence should be rephrased
- l. 354-5: the authors say that BERTbase is more prone to classify tweets as offensive in the Democratic and Republican datasets, but according to Fig.3 it's the opposite (HS tag is preferred, between HS and offensive language, especially in the Republican dataset)
- Sect. 4.3 (and examples in Appendix C): the authors' response to reviewer #3 on this point should be made more explicit in the paper as it would help the reader understand what actually motivated the perturbation analysis and why the authors picked precisely those examples.

Additional comments

Despite the tentative improvements made to the revised version, the paper still needs further adjustments in order to be considered for full acceptance.
Nonetheless, the work is of great interest to the community, as it sheds light on how and to what degree syntax can be helpful in HS detection. Therefore, I highly recommend to the authors that they address reviewers' comments more thoroughly.

---

## Round 0.3 · accepted · Accept

· Academic Editor

Accept

The reviewer's comments have been addressed. Your paper can be published.